# Characteristics and Outcomes of Patients with Acute Coronary Syndrome and COVID-19

**DOI:** 10.3390/jcm11071791

**Published:** 2022-03-24

**Authors:** Aleksandra Milovančev, Milovan Petrović, Višeslav Popadić, Tatjana Miljković, Slobodan Klašnja, Predrag Djuran, Aleksandra Ilić, Mila Kovačević, Anastazija Stojšić Milosavljević, Milica Brajković, Bogdan Crnokrak, Lidija Memon, Ana Milojević, Zoran Todorović, Milenko Čanković, Mirka Lukić Šarkanović, Snežana Bjelić, Snežana Tadić, Aleksandar Redžek, Marija Zdravković

**Affiliations:** 1Faculty of Medicine, University of Novi Sad, 21000 Novi Sad, Serbia; milovan.petrovic@mf.uns.ac.rs (M.P.); tatjana.miljkovic@mf.uns.ac.rs (T.M.); aleksandra.ilic@mf.uns.ac.rs (A.I.); mila.kovacevic@mf.uns.ac.rs (M.K.); anastazija.stojsic@mf.uns.ac.rs (A.S.M.); milenko.cankovic@mf.uns.ac.rs (M.Č.); mirka.lukic-sarkanovic@mf.uns.ac.rs (M.L.Š.); snezana.bjelic@mf.uns.ac.rs (S.B.); snezana.tadic@mf.uns.ac.rs (S.T.); aleksandar.redzek@mf.uns.ac.rs (A.R.); 2Institute of Cardiovascular Diseases of Vojvodina, 21208 Sremska Kamenica, Serbia; 3University Hospital Medical Center Bežanijska Kosa, 11000 Belgrade, Serbia; viseslavpopadic@gmail.com (V.P.); slobodan.klasnja@gmail.com (S.K.); predragdjuran@gmail.com (P.D.); brajkovic.milica@yahoo.com (M.B.); bcrnokrak@yahoo.com (B.C.); memon.lidija@bkosa.edu.rs (L.M.); milojevic.ana@bkosa.edu.rs (A.M.); zoran.tdrvc@gmail.com (Z.T.); sekcija.kardioloska@gmail.com (M.Z.); 4Faculty of Medicine, University of Belgrade, 11000 Belgrade, Serbia

**Keywords:** COVID-19, acute coronary syndrome, mortality, coronavirus, myocardial infarction

## Abstract

Acute coronary syndrome (ACS) in patients with COVID-19 is triggered by various mechanisms and can significantly affect the patient’s further treatment and prognosis. The study aimed to investigate the characteristics, major complications, and predictors of mortality in COVID-19 patients with ACS. All consecutive patients hospitalized from 5 July 2020 to 5 May 2021 for ACS with confirmed SARS-Co-2 were prospectively enrolled and tracked for mortality until 5 June 2021. Data from the electronic records for age and diagnosis, matched non-COVID-19 and COVID-19 ACS group, were extracted and compared. Overall, 83 COVID-19 ACS patients, when compared to 166 non-COVID ACS patients, had significantly more prevalent comorbidities, unfavorable clinical characteristics on admission (acute heart failure 21.7% vs. 6.6%, *p* < 0.01) and higher rates of major complications, 33.7% vs. 16.8%, *p* < 0.01, and intrahospital 30-day mortality, 6.7% vs. 26.5%, *p* < 0.01. The strongest predictors of mortality were aortic regurgitation, HR 9.98, 95% CI 1.88; 52.98, *p* < 0.01, serum creatinine levels, HR 1.03, 95% CI 1.01; 1.04, *p* < 0.01, and respiratory failure therapy, HR 13.05, 95% CI 3.62; 47.01, *p* < 0.01. Concomitant ACS and COVID-19 is linked to underlying comorbidities, adverse presenting features, and poor outcomes. Urgent strategies are needed to improve the outcomes of these patients.

## 1. Introduction

Since the beginning of the Coronavirus disease (COVID-19) pandemic, a significant impact on acute coronary syndrome (ACS) hospitalizations [1] and mortality rates [2] has been reported, even in patients without COVID-19. There is a lack of data regarding clinical characteristics and outcomes, with only a few studies on patients with coexistent COVID-19 and ACS. Early published data advocate poor clinical outcomes [3]. Various reasons for increased myocardial injury in COVID-19 are hypothesized. ACS in the milieu of viral infection and respiratory failure may be related to atherosclerotic plaque rupture, triggered by endothelial cell damage, proinflammatory state, hypoxic injury, coronary spasm [4], and cytokine storm, which may lead to myocardial injury and disbalance between demand and supply [5]. Moreover, severe viral infection can cause a systemic inflammatory response [6] and trigger a procoagulant state which favors coronary artery thrombotic occlusion [7]. All these can lead to type 1 or 2 myocardial infarction. The treatment of these patients is a challenging task, having in mind different complications and potential underlying factors including optimal reperfusion strategy, prolonged ischemic time, and undetermined respiratory status.

The pandemic in Serbia is going through its fourth wave since the first registered case in March 2020. Timely start of vaccination in January 2021 and the opening of three new COVID dedicated hospitals definitely provided better care for the patients and influenced a relatively low mortality rate of around 0.8% [8]. However, the organization of hospitals in terms of primary PCI centers for COVID-19 positive patients with ACS was complex and had the task to shorten the door-to-balloon time as much as possible. For now, every major city in Serbia has one big COVID dedicated hospital with a cath lab, although the impediments of providing proper care still exist, as these patients are additionally endangered by COVID-19 infection and its complications. Serbia provides free urgent medical care for every patient with acute coronary syndrome. Both centers participating in the survey are a part of the ST-elevation myocardial infarction (STEMI) network. The Institute of Cardiovascular Diseases of Vojvodina (ICVDV) is a university regional hub center of the Vojvodina province providing care to 2 million inhabitants (approximately one-quarter of the Serbian population) and performing approximately 1300 primary percutaneous coronary interventions (pPCIs) per year, and the other, the University Hospital Medical Center Bežanijska Kosa (BK), provides medical care in another Serbian region for 600,000 inhabitants. During the pandemic, both hospitals were fully operational, providing complete cardiovascular care for COVID-19 positive patients with acute coronary syndrome.

Our study aimed to investigate characteristics and follow-up of COVID-19 patients with acute coronary syndrome, as well as determine credible predictors of mortality and identify potential significant complications. 

## 2. Materials and Methods

We performed a prospective two-center study aiming to evaluate the characteristics and outcomes of hospitalized consecutive patients aged ≥18 years with ACS and concomitant COVID-19 (over the course of a period from 5 July 2020 to 5 May 2021). 

Consecutive patients who were admitted to the ICVDV and University Hospital Medical Center BK in Serbia were included, in a predefined time frame with the principal diagnosis of ACS, who tested positive for SARS-CoV-2 virus, with an antigenic or real-time polymerase chain reaction (RT-PCR) test on a nasopharyngeal swab sample. University Hospital Medical Center Bežanijska Kosa (Belgrade, Serbia) is a tertiary institution with a cardiology department consisting of 60 hospital beds, 11 intensive care beds, and two 24/7 cath labs. The ICVDV is also a tertiary care institution with a cardiology and cardiosurgery department. The cardiology department has 114 hospital beds, 15 intensive care beds, and three 24/7 cath labs.

Final outcomes were followed until 5 June 2021. A regular follow-up visit was scheduled 30 days after hospitalization at the cardiology department. Later on, patients were followed up on their regular visits (in 90, 180 days), or patients who failed to attend the follow-up visits were contacted and interviewed by telephone. Outcomes like complications and duration of hospitalization were also followed. 

All consecutive patients admitted for ACS, ST-segment elevation myocardial infarction (STEMI), non-ST-segment elevation myocardial infarction (NSTEMI), and unstable angina (UA) in a predefined timeframe were included in the survey. Acute myocardial infarction was defined in accordance with the Fourth Universal Definition of Acute Myocardial Infarction [9]. The data for the historical cohort were retrospectively collected from the electronic medical records of patients hospitalized for ACS in the pre-COVID-19 period (2019) at the ICVDV. Data were collected and analyzed for both groups regarding patient’s risk factors, comorbidities, and clinical findings at admission, length of hospitalization, major complications, echocardiographic findings, laboratory findings, and mortality. Major complications were defined as life-threatening arrhythmias, cardiogenic shock, cardiopulmonary cerebral resuscitation (CPCR), and thrombus in the left ventricle. There were no patients with mechanical complications of MI. Killip classes III and IV were considered acute heart failure. Parameters for the control group were extracted from the electronic records, and data for COVID-19 ACS patients were prospectively collected. For COVID-19 ACS patients, we collected data regarding COVID-19 disease severity, treatment of respiratory failure, and complications including pneumonia or pulmonary embolism. According to the proposed classification from WHO [10], COVID-19 was classified as mild, moderate, and severe. The following assays were used: High Sensitive STAT Troponin I assay and Alere NT-proBNP from Abbott Laboratories, Abbott Park, IL, USA. Commercially available routine assays were used to determine CRP, ALT, AST, LDH, Creatinine, CK and CKMB (Sysmex America Inc., Lincolnshire, IL, USA). 

For the STEMI subgroup of patients, we calculated ischemic times as well, particularly the time from symptom onset until hospital admission, door-to-balloon time (the time from the presentation at the emergency room department until the first balloon inflation during the percutaneous coronary intervention (PCI)), and total ischemic time. These times were calculated for STEMI patients who underwent PCI within 12 h of symptoms onset. There were missing values for CK in 9 patients, for CKMB in 29 patients, for LDH in 17 patients, for hs Troponin I in 48 patients, for CRP in 16 patients, for AST and ALT in 17 patients, for creatinine in 13 patients, for WBC in 13 patients, for Ly% and Ne% in 7 patients, for Hgb in 13 patients, for PLT in 6 patients, for NT-proBNP in 185 patients, for RVSP in 102 patients, for E/e’ in 109 patients, and for EF in 18 patients.

One investigator finally checked the data for missing or contradictory entries and values out of the normal range. The study was conducted according to the principles of the Declaration of Helsinki. The COVID-19 positive patients with ACS that were followed for outcomes signed informed consent. The Institutional Ethics Committee of participating institutions approved the study.

### Statistical Analysis

Continuous variables are presented as the means and standard deviations or median with interquartile ranges 25th and 75th percentile for continuous data. To test the normal distribution, the Kolmogorov-Smirnov test was used. Categorical variables are presented as absolute numbers and percentages. Differences were tested via Student’s paired *t*-test, Wilcoxon and the chi-square test as appropriate. Groups were matched according to a propensity score generated using the following covariates: diagnosis, (STEMI, NSTEMI, and UA) and age. A 1:2 matching was then performed based on this propensity score. Standard differences were calculated for covariates in propensity score matching and used to assess the effectiveness of the match. Cox proportional hazards regression was used to determine independent predictors of mortality among participants, and these were expressed as estimated hazard ratios (HRs) with their corresponding 95% confidence intervals (CIs). Variables that were found to be statistically significant in univariable analysis were used for multivariable model. A backward Cox multivariable regression was used to build the model *p*-values lower than 0.05 were considered statistically significant. The statistical software Statistica (Statistica 13.5, The Ultimate Academic Bundle, StatSoft Europe GmbH, Hamburg, Germany; university license for the University of Novi Sad) was used for all analyses.

## 3. Results

### 3.1. Basic Caracteritics

The study cohort included 83 hospitalized ACS patients with confirmed COVID-19 aged 66.5 ± 11.8 vs. a 166 non-COVID-19 ACS patient control group aged 65.9 ± 10.7 years, *p* = 0.69. Females (31.9%) in the non-COVID group were significantly older than males (68.1%), respectively 71.9 ± 9.5 vs. 63.2 ± 10.2, *p* < 0.01, but not in the COVID ACS group, females (39.8%) aged 68.9 ± 11.4 vs. males (60.2%) 64.9 ± 12 years, *p* = 0.1. The baseline clinical characteristics and comorbidities are listed in Table 1. Of the 83 patients in the COVID-19 ACS group, 77.2% had 1 or more coexisting underlying medical conditions vs. 58.4%, *p* < 0.01, in the non-COVID ACS group. Comorbidities that were significantly prevalent in COVID-19 ACS patients were hypertension (67.5 vs. 53%, *p* = 0.02), diabetes (37.3% vs. 19.9%, *p* < 0.01), chronic kidney disease (12% vs. 1.8%, *p* < 0.01), and prior PCI (9.6% vs. 2.4%, *p* = 0.01).

### 3.2. Clinical Characteristics on Admission

By comparing admission examination parameters between groups, COVID-19 ACS patients had significantly lower median BMI (25.7 (23.5, 28.2) vs. 27.5 (26.2, 28.8), *p* < 0.01), systolic blood pressure (130 (110, 150) vs. 142.5 (126.3, 146.3) mmHg, *p* < 0.01), and diastolic blood pressure (75 (68.7, 81.2) vs. 70 (70, 72.5) mmHg, *p* < 0.01) and higher heart rate (85 (70, 100) vs. 80 (70, 90) bpm, *p* = 0.05) and prevalence of acute heart failure (21.7% vs. 6.6%, *p* < 0.01) (Table 2). The COVID-19 ACS group also exhibited an increased incidence of major complications during hospitalization, 33.7% vs. 16.8%, *p* < 0.01. 

### 3.3. Laboratory Findings

Complete blood count revealed that COVID-19 ACS patients had significantly higher median levels of Ne%, 77.1 (66.2, 83.1) vs. 75.5 (68.7, 82.5), *p* < 0.01, and lower Ly%, 12.9 (8.8, 24.3) vs. 17 (14.4, 23.5), *p* < 0.01. The marker of inflammation, C reactive protein, was also higher in the COVID-19 group, median 78 (5.5, 208.5) vs. 22.6 (15.4, 61.8), *p* = 0.02, as well as a tissue damage biomarker median, LDH 563 (446.5, 812) vs. 292 (275.7, 573.5), *p* = 0.01 (Table 3).

### 3.4. Echocardiography

Mean EF in COVID-19 ACS patients was 42.2 ± 11.2 vs. 45.8 ± 7.7 in non-COVID-19 (Table 4). Reduced EF below 40% was reported in 25.3% of COVID-19 ACS vs. 19.8%, *p* = 0.3, of non-COVID ACS patients. Patients with COVID-19 and ACS tend to have more severe valvular insufficiency. Moderate MR was significantly more prevalent in the COVID-19 ACS group, 31.3% vs. 10.9%, *p* < 0.01, as well as moderate TR, 21.7% vs. 8.5%, *p* ≤ 0.01, and severe TR, 2.4% vs. 0%, *p* = 0.01. 

### 3.5. STEMI

Concerning the STEMI subgroup of patients, early presentation (≤6 h from symptom onset) was significantly more prevalent in non-COVID ACS patients, 77.9% vs. 59.6%, *p* < 0.01. While very late STEMI presentation (>12 h from symptom onset) was more often seen in COVID-19 ACS patients, 24.5% vs. 10.8%, *p* < 0.01 (Figure 1). 

Median time from symptoms onset until hospital admission was significantly prolonged in the COVID-19 ACS group (285 (135, 360) vs. 180 (120, 240) min, *p* = 0.02) as well as median door-to-balloon time (54 (45–78) vs. 47 (33–61) min, *p* < 0.01) and total ischemic time (322.5 (199.0–437.5) vs. 218 (169–312) min, *p* < 0.01).

We observed comparable PCI rates among STEMI patients (non-COVID-19 96.4% vs. COVID-19 ACS 91.9%, *p* = 0.21) between groups, while NSTEMI PCI rates were significantly lower in the COVID-19 ACS group, 23.8% vs. 76.2%, p < 0.01. For the COVID-19 STEMI subgroup, only 1.7% had non-obstructive coronary artery disease; the left main was judged the culprit lesion in 3.4%, in 37.3% the left anterior descending artery (LAD), in 5.1% the left circumflex (LCX), in 44% the right coronary artery (RCA), and 10.1% patients had multivessel coronary artery disease (CAD). 

For the STEMI subgroup of patients, we observed significantly higher values of cardiac biomarkers in the COVID-19 group, especially hs troponin I, respectively, 10,327 (941.4, 40,000) vs. 838.8 (140.4, 13,682), *p* = 0.01, and NT-pro-BNP 5031 (736, 13,341) vs. 193 (81, 2542), *p* < 0.01 (Table 5). 

### 3.6. COVID-19 and ACS

The prevalence of patients who tested positive before admission for ACS was 55.4%, while in 43.4%, the diagnosis of ACS was found first. 

Mild COVID-19 disease was found in 34 (41%) patients, moderate in 24 (28.9%), and severe in 25 (30%). Of twenty-five severe COVID-19 patients, seven were treated using non-invasive ventilation (NIV), and 18 via invasive mechanical ventilation (IMV). Pneumonia as a complication of COVID-19 was found in 59 (71%) hospitalized ACS patients. 

Three patients with ACS had a pulmonary embolism (PE), while no PE was found in the control group.

### 3.7. Mortality

The median follow-up time was 201.5 (20, 201.5) days. The median time from diagnosis of COVID-19 until death was 9 (3, 18.5) days. Overall, mortality was 28.9% in COVID-19 ACS patients, and 30 days mortality was 26.5%. Two additional deaths were found during follow-up; one patient died after 39 days and one after 75 days, both due to respiratory failure. 

Intrahospital 30-day mortality was significantly increased from 6.6% in non-COVID-19 to 26.5% in COVID-19 ACS patients, *p* < 0.01, for STEMI patients 5.1% vs. 28.8%, *p* < 0.01, and for NSTEMI 11.1% vs. 22.2%, *p* = 0.2. 

Univariable Cox regression analysis (Table 6) identified that significant predictors of mortality were age, AR, TR, MR, EF, HR, Killip, door-to-balloon time, history of chronic kidney disease, serum creatinine levels, Ly%, Ne%, and respiratory failure therapy. 

Multivariable backward stepwise final Cox model predictors of mortality were aortic regurgitation HR 9.98 (1.880; 52.988), *p* = 0.007, serum creatinine levels, HR 1.03, 95% CI 1.007; 1.047, *p* = 0.007, and respiratory failure therapy, HR 13.05, 95% CI 3.622; 47.015, *p* = 0.001. 

## 4. Discussion

In our study, we report significantly increased complications and mortality rates in COVID-19 ACS patients. This is the first study with a follow-up of patients after ACS and COVID-19 hospitalization. 

Individuals with COVID-19 and ACS were more prevalent in underlying comorbidities, including a history of hypertension, diabetes, kidney disease, and prior PCI. Earlier studies reported that 63% of hospitalized COVID-19 patients had hypertension, diabetes, cardiovascular diseases, and malignancy as the most common coexisting conditions [11]. A case series of 28 COVID-19 STEMI patients reported results similar to ours where 71.4% had arterial hypertension, 32.1% diabetes, 28.6% chronic kidney disease, and 10.7% previous MI [12]. 

Patients with ACS and COVID-19 on hospital admission presented more often with lower blood pressure, higher heart rate, and higher incidence of acute heart failure and major complications. Laboratory findings revealed significant differences in inflammatory and heart failure biomarkers in COVID ACS patients. Rashid et al. [12] reported similar results in the group of COVID-19 ACS patients that exhibited an increased incidence of in-hospital cardiac arrest and were more likely to present with pulmonary edema or cardiogenic shock.

Patients with STEMI and COVID-19 tend to come later as we observed significant delays from symptoms onset until hospitalization, time to revascularization, and total ischemic time. This for sure resulted in higher troponin and NT-proBNP values due to increased tissue and myocardial injury. 

Earlier reports [3,12] speculated that COVID-19 ACS patients had a higher incidence of non-obstructive CAD. In our study, PCI rates among STEMI patients were 91.9%, and the prevalence of non-obstructive CAD was 1.7%, indicating that most patients had obstructive CAD. At the same time, NSTEMI PCI rates were significantly lower in the COVID-19 ACS group at 23.8% vs. 76.2%, <0.01. Later reports on larger samples (156 [13] and 91 [14] COVID-19 ACS patients) revealed a low prevalence of non-obstructive CAD, in line with our results.

Almost one-third of the patients died, with the STEMI patients exhibiting higher mortality rates than NSTEMI. High mortality rates of 39.3% of COVID-19 patients with STEMI were also reported by Stefanini et al. [12], with 41.9% reported by Rashid [13], 72% by Bangalore [3], and 26% by Hamadeh [15]. A higher mortality rate is a result of severe disease, increased oxygen demand, and a procoagulant state. The therapy used in the treatment of COVID-19 could also have a significant impact on cardiovascular function, leading to a more deteriorating clinical course in these patients. In our study, independent predictors of mortality were aortic regurgitation, serum creatinine levels, and respiratory failure therapy. The need for respiratory failure therapy in patients with ACS and COVID-19 (conventional oxygen therapy, NIV, IMV) was the strongest predictor of mortality. Patients whose respiratory failure was treated with non-invasive ventilation had 13 times increased risk for death than those on conventional oxygen therapy. In the study that included 32 COVID-19 positive patients with acute myocardial infarction, overall mortality was 25%. Patients with ARDS on mechanical ventilation had dramatically higher mortality than those without ARDS (83% vs. 12%, *p* = 0.002) [16] and likewise in our study. Rasid et al. found that COVID-19 ACS patients had significantly higher in-hospital and 30-day mortality with identified elevated creatinine, peak troponin, heart rate, left ventricular systolic dysfunction, and use of ACE inhibitor or ARBs as independent risk factors for 30-day mortality [13]. The causes for dramatically increased mortality in patients with COVID and ACS may be various. Direct toxicity of the virus inside the myocyte through angiotensin-converting enzyme-2 membrane receptors may cause additional myocardial inflammation and injury. In STEMI patients in our report, significantly higher hs Troponin levels in the COVID group may support this hypothesis. Elevated CRP in ACS patients is correlated with increased 14-day mortality [17]. Increased supply-demand mismatch of oxygen in the environment of proinflammatory state and hypoxemia-mediated processes due to acute respiratory distress syndrome may increase the risk of adverse outcomes and be one of the reasons for significantly higher mortality rates in these patients. Late STEMI presentations have been widely reported during the COVID pandemic [2,18]. Deferring urgent medical care and late presentation of STEMI patients have resulted in later treatment, which inevitably worsens outcomes. Uncovering the reasons and predictors of increased mortality in COVID ACS patients for sure can improve timely diagnostics, treatment, and outcomes. The present study has several limitations. Mainly, the sample size was relatively small, although the number of participants and statistical results were consistent with previous studies regarding patients with acute coronary syndrome and COVID-19 disease. Hospital centers performing primary PCI covered different territories and a number of citizens, potentially affecting ischemic time due to potentially different STEMI network protocols being conducted. Finally, patients in one center were transported to another care unit specializing in COVID-19 treatment after getting a positive SARS-CoV-2 test, while patients in the other center were treated for ACS and COVID-19 in the same intensive care unit.

## 5. Conclusions

Acute coronary syndrome associated with concomitant COVID-19 infection is linked with underlying comorbidities, adverse presenting features, and poor clinical outcomes. Urgent strategies and further larger registries are necessary to improve the diagnostics and treatment and decrease the mortality of these patients. 

## Figures and Tables

**Figure 1 jcm-11-01791-f001:**
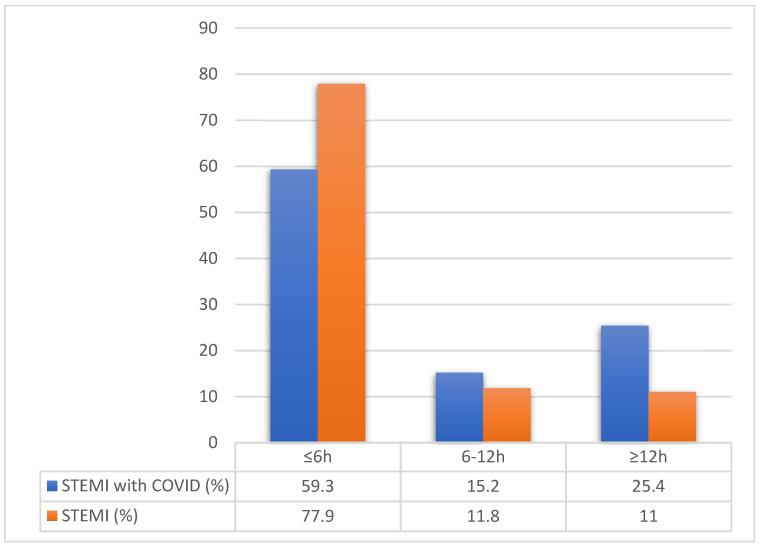
Time from symptoms onset until hospitalization compared between STEMI groups. Legend: STEMI—ST elevated myocardial infarction, COVID—Coronavirus disease.

**Table 1 jcm-11-01791-t001:** Baseline characteristics, comorbidities, and risk factors.

ACS Patient Characteristics	Standardized Differences after Matching	ACS with COVID-19 (*n* = 83) *n* (%), Mean ± SD	ACS (*n* = 166) *n* (%), Mean ± SD	*p*-Value
Gender (M)		60.2%	68.1%	0.26
Age	0.05	66.53 ± 11.86	65.95 ± 10.75	0.69
STEMI	0.00	59 (71.1%)	118 (71%)	0.56
NSTEMI	0.00	21 (25.3%)	42 (25.3%)	0.55
AP	0.00	3 (3.6%)	6 (3.7%)	0.62
History of hypertension		56 (67.5%)	88 (53%)	0.02
History of diabetes		31 (37.3%)	33 (19.9%)	<0.01
History of hyperlipidemia		20 (24.1%)	28 (16.9%)	0.11
History of chronic kidney disease		10 (12%)	3 (1.8%)	<0.01
Family history of CVD		21 (25.3%)	35 (21.1%)	0.27
Smoking history		26 (31.3%)	56 (34.1%)	0.38
COPD		3 (3.6%)	4 (2.4%)	0.42
Malignancy		4 (4.8%)	3 (1.8%)	0.17
Peptic ulcer disease		2 (2.4%)	3 (1.8%)	0.17
Prior stroke		3 (3.6%)	4 (2.4%)	0.42
Prior MI		9 (10.8%)	12 (7.2%)	0.23
Prior PCI		8 (9.6%)	4 (2.4%)	0.01
Prior CABG		3 (3.6%)	3 (1.8%)	0.3

Legend: ACS—acute coronary syndrome, CABG—coronary artery bypass grafting, COPD—chronic obstructive pulmonary disease, CVD—cardiovascular disease, NSTEMI—non-ST-segment elevated myocardial infarction, MI—myocardial infarction, PCI—percutaneous coronary intervention, STEMI—ST-segment elevated myocardial infarction, UA—unstable angina.

**Table 2 jcm-11-01791-t002:** Patients’ clinical examination characteristics on admission and major complications between groups.

Parameter	ACS with COVID-19 (*n* = 83) Median (Q1, Q3) *n* (%)	ACS (*n* = 166) Median (Q1, Q3) *n* (%)	*p*-Value
BMI (kg/m^2^)	25.7 (23.5, 28.2)	27.5 (26.2, 28.8)	<0.01
Blood pressure systolic (mmHg)	130 (110, 150)	142.5 (126.3, 146.3)	<0.01
Blood pressure diastolic (mmHg)	75 (68.7, 81.2)	70 (70, 72.5)	<0.01
Heart rate bpm (bpm)	85 (70, 100)	80 (70, 90)	0.05
Killip I and II	65 (78.3%)	146 (87.9%)	0.04
Killip III and IV	28 (21.7%)	11 (6.6%)	<0.01
Major complications (%)			
VT/VF	7 (8.4%)	7 (4.2%)	0.17
Shock	12 (14.4%)	3 (1.8%)	<0.01
CPCR	5 (6%)	15 (9%)	0.41
Thrombus in left ventricle	0	1 (0.6%)	0.48
Complete AV block	4 (4.8%)	2 (1.2%)	0.08
All	28 (33.7%)	28 (16.8%)	<0.01

There was a statistically significant difference in the median length of hospitalization for COVID-19 patients with ACS 2 (1–6) vs. non-COVID ACS patients 5 (3–7) days, *p* = 0.001. At the time of the COVID-19 pandemic, after hospitalizations at cardiology departments, patients were transferred to a Infectious Disease Clinic for further treatment of COVID-19.

**Table 3 jcm-11-01791-t003:** Laboratory findings at admission to hospital.

Parameter	ACS with COVID-19 (*n* = 83) Median (Q1, Q3)	ACS (*n* = 166) Median (Q1, Q3)	*p*-Value
WBC × 10^9^/L	10.3 (7.6, 13.4)	11.6 (10.9, 14,5)	0.04
Ne%	77.1 (66.2, 83.1)	75.5 (68.7, 82.5)	<0.01
Ly%	12.9 (8.8, 24.3)	17 (14.4, 23.5)	<0.01
Hgb g/L	133 (116, 147)	119 (118, 126)	0.11
PLT × 10^9^/L	232 (183, 284)	256 (223, 272)	0.77
CRP mg/L	78 (5.5, 208.5)	22.6 (15.4, 61.8)	0.02
LDH U/L	563 (446.5, 812)	292 (275.7, 573.5)	0.01
Creatinine µmol/L	93 (79, 121)	87 (76.2, 94)	0.69
CK U/L	268 (136, 657)	159 (126.7, 238)	0.60
CKMB U/L	46 (29, 102)	62 (49.2, 64)	0.66
hs Troponin I ng/L	4063 (337, 35,198)	1712.8 (1436.7, 4876.4)	0.37
NT-proBNP pg/mL	4972 (973, 10,364.65)	1136 (568, 13,068)	0.17

Legend: ACS—Acute coronary syndrome, CRP—C reactive protein, CK—creatine kinase, CKMB—creatine kinase-MB, Hgb—hemoglobin, LDH—lactate dehydrogenase, Ly—lymphocytes, Ne—neutrophils, NT-proBNP—N-terminal pro b-type natriuretic peptide, PLT—platelets, WBC—white blood cells count.

**Table 4 jcm-11-01791-t004:** Echocardiographic findings.

Parameter	ACS with COVID-19 (*n* = 83) Mean ± SD %	ACS (*n* = 166) %	*p*-Value
EF	42.2 ± 11.2	45.8 ± 7.7	0.92
E/e’	10.8 ± 3.7	12 ± 3.7	0.62
RVSP	35 ± 9	33.9 ± 10.6	0.23
MR mild	39 (47%)	104 (62.6%)	0.01
MR moderate	26 (31.3%)	18 (10.9%)	<0.01
MR severe	4 (4.8%)	3 (1.8%)	0.1
TR mild	23 (27.7%)	122 (73.5%)	<0.01
TR moderate	18 (21.7%)	14 (8.5%)	<0.01
TR severe	2 (2.4%)	0	0.01
AR mild	25 (30.2%)	116 (69.8%)	<0.01
AR moderate	3 (3.6%)	1 (0.6%)	0.07

Legend: ACS—acute coronary syndrome, AR—aortic regurgitation, E/e´—Ratio of the maximum velocity of the E-wave of mitral valve inflow by the maximal velocity of tissue Doppler e’, EF—ejection fraction, MR—mitral regurgitation, RVSP—right ventricular systolic pressure, TR—tricuspid regurgitation.

**Table 5 jcm-11-01791-t005:** Laboratory findings for STEMI patients.

Parameter	STEMI with COVID-19 (*n* = 59) Median (Q1, Q3)	STEMI (*n* = 118) Median (Q1, Q3)	*p*-Value
LDH U/L	349 (208.5, 539.3)	227 (181, 382)	0.01
AST U/L	49 (30, 92)	39 (27, 97)	0.02
ALT U/L	28 (25, 62)	41 (25, 60)	0.72
CK U/L	418.5 (172.5, 967.3)	208 (122, 650)	0.04
CKMB U/L	53.5 (37, 128.5)	42 (28, 101)	0.08
hs Troponin I ng/L	10,327 (941.4, 40,000)	838.8 (140.4, 13,682)	0.01
NT-proBNP pg/ml	5031 (736, 13,341)	193 (81, 2542)	<0.01

Legend: AST—aspartate transaminase, ALT—alanine transaminase, CK—creatine kinase, CKMB—creatine kinase-MB, LDH—lactate dehydrogenase, NT-proBNP—N-terminal pro b-type natriuretic peptide.

**Table 6 jcm-11-01791-t006:** Independent predictors of mortality in COVID-19 ACS patients.

Univariable			
Variable	HR (Exp B)	95% CI (Lower; Upper)	*p*
Age	1.05	1.01; 1.08	<0.01
Aortic regurgitation	2.67	1.41; 5.03	<0.01
Tricuspid regurgitation	2.55	1.49; 4.33	<0.01
Mitral regurgitation	3.95	1.96; 7.95	<0.01
EF	0.94	0.90; 0.97	<0.01
HR bpm	1.04	1.02; 1.06	<0.01
Killip	2.23	1.61; 3.09	<0.01
DBT	1.01	1.00; 1.01	<0.01
Chronic kidney disease	3.82	1.49; 9.76	<0.01
Creatinine	1.00	1.001; 1	<0.01
Ly%	0.89	0.83; 0.95	<0.01
Ne%	1.01	1.04; 1.14	<0.01
Respiratory failure therapy	5.64	3.28; 9.68	<0.01

Legend: bpm—beat per minute, CI—confidence interval, DBT—door-to-balloon time, EF—ejection fraction, HR—heart rate, Ly—lymphocyte, Ne—neutrophils.

## Data Availability

The data presented in this study are available on request from the corresponding author.

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
