# Peer review of "Characteristics and Outcomes of Patients with Acute Coronary Syndrome and COVID-19"

_jcm, 2022, doi:10.3390/jcm11071791_

Round 1

Reviewer 1 Report

Thank you for youre revised manuscript. Here are my comments. 1.All p-value are consistance, which is good. Why did the result of table2 and table 3 for conitunous variable change from first manuscript? Which values should we trust? 2. Table1 must be the result after propensity score matching. Please indicate standard mean difference to show the dataset is well-matched. Also, please include original dataset in either Table1 or Supplyment table to show the background of this dataset. Make sure to state about standard mean difference in statistical method. 3.Table 6 does not follow the statistical method. Authors mentioned "Variables that were found statistically significant in univariable models were used for multivariate model". But, Age was not used in multivariate analysis. Is this still step-wise method? Or just multivariate analaysis? Annotation in Table6 still showed step-wise, which is confusing. 4.Authors mentioned that they discussed about table 6 in discussion. I believed they mentioned in line 320-324. But, this is just "summary" of Table6, not discussion. Please compare with other research and mention the difference or new-insight in this research. Otherwises, as I mentioned, Table 6 may be omitted.Overall, discussion section sounds like just "summmary" to me. 5.Please re-consider about multivariate and multivariable. I believe authors should state multivariable. Reference: Hidalgo B, Goodman M. Multivariate or multivariable regression? Am J Public Health. 2013 Jan;103(1):39-40

Author Response

Thank you for your revised manuscript. Here are my comments.

1.All p-value are consistance, which is good.

Thank you for your comment.

Why did the result of table2 and table 3 for conitunous variable change from first manuscript? Which values should we trust?

The Reviewer 2 suggested that a new statistical analysis should be done for the paired data to improve the results, so I calculated again as the reviewer asked.  Though there are no big differences, the p values that were significant remained significant except Hgb and NTproBNP.

  1. Table1 must be the result after propensity score matching. Please indicate standard mean difference to show the dataset is well-matched. Also, please include original dataset in either Table1 or Supplyment table to show the background of this dataset. Make sure to state about the standard mean difference in statistical method.

The patients were matched by diagnosis and age, standard difference for the age and diagnosis is calculated and presented in Table 1. The statement is added in the statistical method also. Unfortunately. The original dataset can not be presented, due to the policy of the two Institutions.

3.Table 6 does not follow the statistical method. Authors mentioned "Variables that were found statistically significant in univariable models were used for multivariate model". But, Age was not used in multivariate analysis. Is this still step-wise method? Or just multivariate analaysis? Annotation in Table6 still showed step-wise, which is confusing.

Variables that were found statistically significant in univariable models were used for the multivariable model, but some of them were not found to be statistically significant in the final model (for example age, p = 0.176). This is a multivariable analysis. The annotation and statistical method is corrected.

 4.Authors mentioned that they discussed about table 6 in discussion. I believed they mentioned in line 320-324. But, this is just "summary" of Table6, not discussion. Please compare with other research and mention the difference or new-insight in this research. Otherwises, as I mentioned, Table 6 may be omitted.Overall, discussion section sounds like just "summmary" to me.

The discussion section is now corrected

5.Please re-consider about multivariate and multivariable. I believe authors should state multivariable. Reference: Hidalgo B, Goodman M. Multivariate or multivariable regression? Am J Public Health. 2013 Jan;103(1):39-40

Thank you for your comment. Multivariable is now used throughout the whole manuscript.

Reviewer 2 Report

I greatly appreciated the authors' work in improving the manuscript. However, authors should make changes to the text.

Authors should indicate which kits are used in measuring the various biomarkers. For example,  Have you used Abbott hs-ctnI kit for troponin I measurement?

These informations are of the utmost importance.

Author Response

I greatly appreciated the authors' work in improving the manuscript. However, authors should make changes to the text.

Thank you for your comment.

Authors should indicate which kits are used in measuring the various biomarkers. For example,  Have you used Abbott hs-ctnI kit for troponin I measurement?

These informations are of the utmost importance.

The following assays were used: High Sensitive STAT Troponin I assay and Alere NT-proBNP from Abbott Laboratories, Abbott Park, IL, USA. Commercially available routine assays from Simens Medical solutions USA were used to determine CRP, ALT, AST, LDH, Creatinine, CK and CKMB.

This section is added in the Material and methods

Round 2

Reviewer 1 Report

Standard mean difference can only show after matching since we are not allowed to observe before the matching according to the authors.

Table6 still shows step-wise method to me. Since the method states "Variables that were found statistically significant univariable models were used for multivariable model", all variables that may not be siginificant in multivariable model should state in the table. (This is third time mentioned).

Author Response

Respected reviewer,

I want to thank you for your quick answers and great effort to improve the manuscript.

Standard mean difference can only show after matching since we are not allowed to observe before the matching according to the authors.

Standard mean difference showed after matching in table one

Table6 still shows step-wise method to me. Since the method states "Variables that were found statistically significant univariable models were used for multivariable model", all variables that may not be siginificant in multivariable model should state in the table. (This is third time mentioned).

You are right. I talked to my statistician again, a backward Stepwise Cox multivariable regression was used.  Table 6 is corrected to avoid confusion.

 Thank you again

Reviewer 2 Report

I greatly appreciated the authors' work in improving the manuscript.

Author Response

Thank you so much for your comments and help